# Diverse ancestral representation improves genetic intolerance metrics

Alexander L. Han [1,2,10], Chloe F. Sands[1,2,10], Dorota Matelska[3], Jessica C. Butts[4,5], Vida Ravanmehr[1,2], Fengyuan Hu[3], Esmeralda Villavicencio Gonzalez[2,6], Nicholas Katsanis[7], Carlos D. Bustamante[7], Quanli Wang[8], Slavé Petrovski [3,9] ✉, Dimitrios Vitsios [3] & Ryan S. Dhindsa [1,2,6] ✉

The unprecedented scale of genomic databases has revolutionized our ability to identify regions in the human genome intolerant to variation—regions often implicated in disease. However, these datasets remain constrained by limited ancestral diversity. Here, we analyze whole-exome sequencing data from 460,551 UK Biobank and 125,748 Genome Aggregation Database (gnomAD) participants across multiple ancestries to test several key intolerance metrics, including the Residual Variance Intolerance Score (RVIS), Missense Tolerance Ratio (MTR), and Loss-of-Function Observed/Expected ratio (LOF O/E). We demonstrate that increasing ancestral representation, rather than sample size alone, critically drives their performance. Scores trained on variation observed in African and Admixed American ancestral groups show higher resolution in detecting haploinsufficient and neurodevelopmental disease risk genes compared to scores trained on European ancestry groups. Most strikingly, MTR trained on 43,000 multi-ancestry exomes demonstrates greater predictive power than when trained on a nearly 10-fold larger dataset of 440,000 non-Finnish European exomes. We further find that European ancestry group-based scores are likely approaching saturation. These findings highlight the need for enhanced population representation in genomic resources to fully realize the potential of precision medicine and drug discovery. Ancestry group-specific scores are publicly available through an interactive portal: http://intolerance.public.cgr.astrazeneca.com/.

The emergence of population-scale sequencing datasets has enabled the identification of regions of the human genome that are intolerant to functional variation due to negative selection[1–5]. Broadly, intolerance scores provide a quantitative measure of the depletion of variants in a gene compared to a null expectation in the general population. We and others have developed intolerance metrics that quantify the intolerance of genes, genic sub-regions, and non-coding regions[2,6–12]. These scores have become a cornerstone in prioritizing genetic

[1]Department of Pathology and Immunology, Baylor College of Medicine, Houston, TX, USA. [2]Jan and Dan Duncan Neurological Research Institute, Texas Children's Hospital, Houston, TX, USA. [3]Centre for Genomics Research, Discovery Sciences, BioPharmaceuticals R&D, AstraZeneca, Cambridge, UK. [4]Department of Bioengineering, George R. Brown School of Engineering, Rice University, Houston, TX, USA. [5]Rice Neuroengineering Initiative, George R. Brown School of Engineering, Rice University, Houston, TX, USA. [6]Department of Molecular and Human Genetics, Baylor College of Medicine, Houston, TX, USA. [7]Galatea Bio, Inc, Miami, FL, USA. [8]Centre for Genomics Research, Discovery Sciences, BioPharmaceuticals R&D, AstraZeneca, Waltham, MA, USA. [9]Department of Medicine, Austin Health, University of Melbourne, Melbourne, VIC, Australia. [10]These authors contributed equally: Alexander L. Han, Chloe F. Sands. ✉e-mail: slav.petrovski@astrazeneca.com; ryan.dhindsa@bcm.edu

**Table 1 | Tally of common (MAF > 0.05%) and rare protein-coding variants observed in the gnomAD ancestral groups**

| Variant Type | AFR (n = 8128) | AMR (n = 17,296) | SAS (n = 15,308) | EAS (n = 9197) | ASJ (n = 5040) | NFE (n = 56,885) | FIN (n = 10,824) |
|---|---|---|---|---|---|---|---|
| Common Missense | 141,538 | 105,985 | 103,281 | 93,436 | 81,093 | 79,200 | 70,438 |
| Common PTVs | 6694 | 5451 | 5306 | 5321 | 4781 | 4447 | 4263 |
| Common Synonymous | 115,737 | 84,509 | 78,068 | 66,795 | 60,801 | 59,348 | 50,748 |
| Rare Missense | 652,095 | 949,110 | 1,038,844 | 619,444 | 103,944 | 2,384,930 | 212,055 |
| Rare PTV | 49,241 | 74,192 | 79,637 | 48,861 | 11,469 | 219,425 | 22,077 |
| Rare Synonymous | 345,648 | 491,853 | 536,568 | 317,486 | 50,053 | 1,143,528 | 99,547 |

**Table 2 | Tally of common (MAF > 0.05%) and rare protein-coding variants observed in the UKB ancestral groups**

| Variant Type | AFR (n = 8701) | SAS (n = 9217) | EAS (n = 2150) | ASJ (n = 2671) | NFE (n = 20k) | NFE (n = 43k) | NFE (n = 440k) |
|---|---|---|---|---|---|---|---|
| Common Missense | 150,816 | 112,862 | 99,032 | 88,248 | 78,198 | 78,736 | 78,272 |
| Common PTVs | 7647 | 6278 | 5718 | 5474 | 4650 | 4630 | 4684 |
| Common Synonymous | 125,683 | 84,416 | 72,860 | 65,668 | 59,161 | 59,429 | 59,271 |
| Rare Missense | 859,599 | 854,077 | 288,419 | 58,609 | 1,175,279 | 1,859,809 | 6,023,520 |
| Rare PTV | 69,882 | 70,874 | 22,267 | 6,639 | 112,946 | 189,338 | 772,007 |
| Rare Synonymous | 456,784 | 445,392 | 155,012 | 29,249 | 576,402 | 886,465 | 2,633,791 |

variants in diagnostic settings, discovering genes and genomic regions underlying human traits, and predicting drug targets[13].

Gene-level intolerance metrics provide a valuable resource for studying the functional significance of human genes. For example, we previously introduced the Residual Variance Intolerance Score (RVIS), which uses standing variation in the human population to rank genes based on their tolerance to common functional variation (i.e., missense and protein-truncating variants)[1]. Other gene-level scores focus on more specific classes of mutations, such as LOF-FDR and LOEUF (loss-of-function intolerance) and missense Z (missense intolerance) (see Supplementary Table 1 for a glossary of abbreviations)[4,5,14]. While gene-level metrics have proven extremely useful, certain regions of a gene can be more intolerant than other regions within the same gene, motivating the development of intragenic intolerance metrics. One example is the sliding-window missense tolerance ratio (MTR), which identifies genic sub-regions that are intolerant to missense variation using a sliding window[9].

As population-level sequencing datasets continue to expand in sample size, the genetic variants observed in these datasets also increase in number. Such an increase in the number of observed variants should, in theory, improve the performance of intolerance metrics. However, current datasets remain disproportionately enriched for individuals of Northern European ancestry. Given that genetic variants and their frequencies differ dramatically across different populations[15], we reasoned that the underrepresentation of global ancestries in these datasets not only exacerbates health inequities[16,17], but could also limit the resolution of genic intolerance metrics.

In this work, we explored the extent to which ancestral diversity impacts the performance of genic and sub-genic intolerance metrics. Using population-level datasets from the Genome Aggregation Database (gnomAD) and the UK Biobank (UKB), we demonstrate that broad ancestral representation increases the performance of several intolerance metrics in predicting neurodevelopmental disease and haploinsufficient genes. Notably, RVIS and missense tolerance metrics derived from African ancestry cohorts outperformed those computed from predominantly European individuals. These findings underscore the importance of broadening global ancestry representation to achieve

equitable genomic research, enhance target discovery, and deepen our understanding of human biology.

## Results

### gnomAD and UKB cohort characteristics

We leveraged large collections of exome sequence data from gnomAD (v2.1) and the UKB to measure the impact of ancestral diversity on genic and sub-genic intolerance metrics. We used previously existing ancestral group classifications for both these resources[14,17]. Briefly, these classifications were derived using Principal Component Analysis methods to identify clusters of genetic similarity, which were then labeled based on established reference panels (see Methods).

The gnomAD v2 dataset includes aggregated allele frequency data from 125,748 exomes, among which we analyzed ancestry group-specific allele frequencies from African (n = 8128), South Asian (n = 15,308), Latino (n = 17,296), East Asian (n = 9197), Ashkenazi Jewish (n = 5040), non-Finnish European (n = 56,885), and Finnish populations (n = 10,824). In the UKB, we analyzed exome data from 460,551 individuals. Of these, 437,812 (95.06%) participants were of non-Finnish European ancestry, 8701 (1.89%) were of African ancestry, 2671 (0.58%) were of Ashkenazi Jewish ancestry, 9217 (2.00%) were of South Asian ancestry, and 2150 (0.47%) were of East Asian ancestry.

As expected, the African ancestry cohorts (AFR) exhibited the most genetic diversity, relative to the current reference genome (hg38), followed by the Admixed American (AMR) and South Asian (SAS) cohorts. This difference was particularly apparent for common (MAF > 0.05%) functional variants (Tables 1, 2). For example, in gnomAD there was a 1.8-fold enrichment of common missense variants in the AFR cohort (n = 141,538 variants among 8,128 individuals) compared to only the NFE cohort (n = 79,200 variants in 56,885 individuals). Similarly, there was a roughly 1.6-fold enrichment of common PTVs in the AFR subset versus the NFE subset (Tables 1, 2). The Finnish European group had the fewest common missense variants and PTVs, in part attributable to the founding bottleneck event that occurred roughly 120 generations ago, which reduced the effective gene pool and thus set their genetic history apart from other European populations[18,19].

At current sample sizes, it appears that we have saturated the number of common (MAF > 0.05%) functional variants observed in

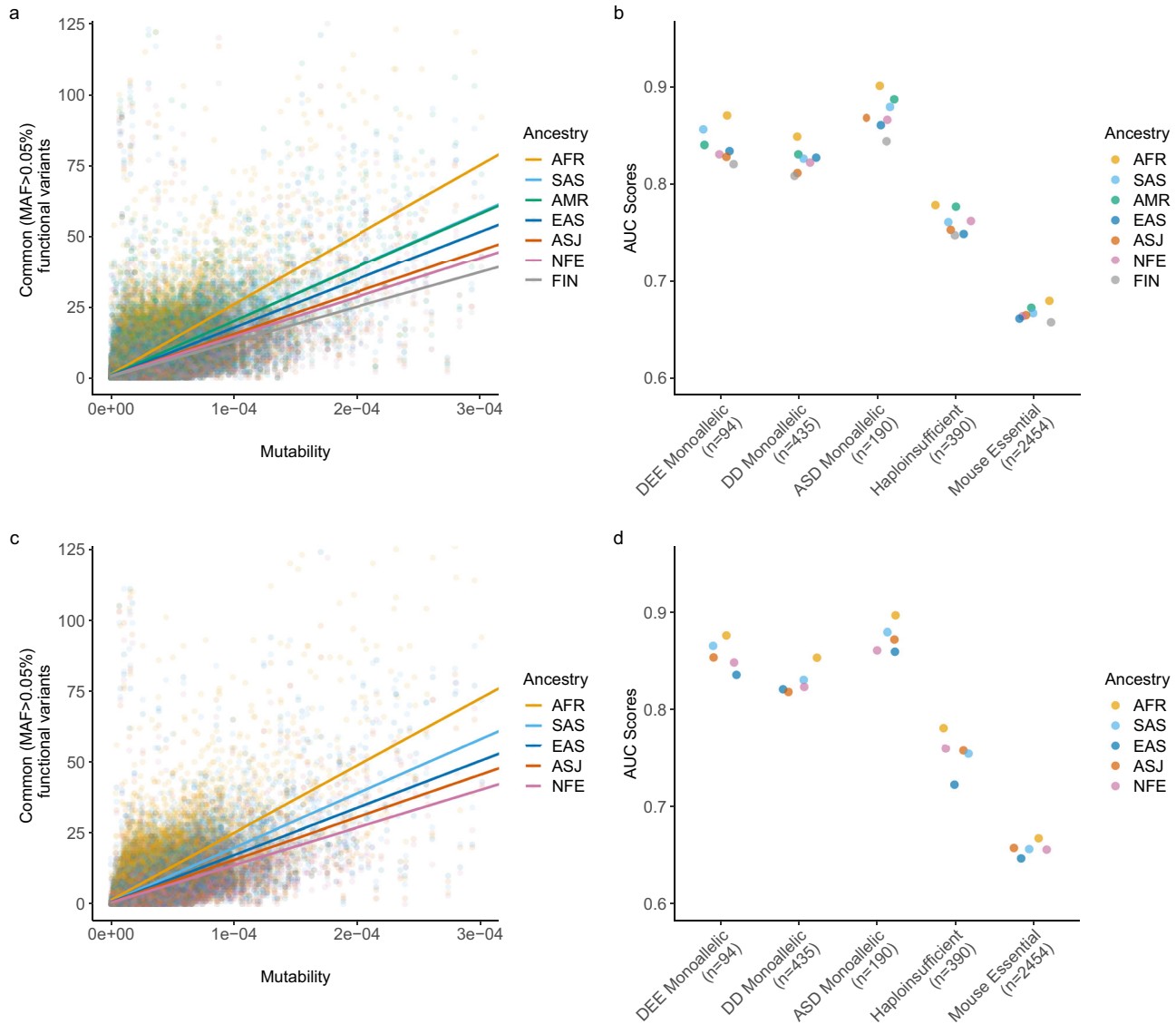

**Fig. 1 | Ancestry group-specific RVIS in gnomAD and UKB. a** Scatter plot illustrating the regression of common (MAF > 0.05%) functional variants on mutability for seven different ancestry groups in gnomAD. The X- and Y-axes are capped to remove outlier genes. **b** AUC-ROC values of the ancestry group-specific RVIS in gnomAD for predicting five gene lists. **c** Regression lines of common (MAF > 0.05%) functional variants versus mutability for five different ancestry groups in the UKB. **d** AUC-ROC values of the ancestry group-specific RVIS in the UKB for predicting the following five gene lists. AFR African, AMR Admixed American, ASJ Ashkenazi Jewish, EAS East Asian, FIN Finnish, NFE non-Finnish European, SAS South Asian, DEE developmental and epileptic encephalopathy, DD developmental delay, ASD autism spectrum disorder.

non-Finnish Europeans. To demonstrate this, we created three subsets of NFE UKB participants: one with 20,000 individuals, one with 43,000, and one with 440,000 individuals. The number of common variants was stable across these three cohorts (Table 2). However, the number of rare variants increased proportionally with sample size, apparent in the NFE subsets as well as the other major ancestral groups (Tables 1, 2). Over 50% of the newly added variants were singletons (Supplementary Fig. 1). Collectively, these results demonstrate that enhancing diversity, in addition to sample size, is critical for better representing standing variation in the global human population.

**Deriving ancestry group-specific RVIS**

Given the significant differences in the amount of functional variation observed between populations, we hypothesized that lack of genetic diversity could limit the resolution of intolerance metrics. We first investigated this by creating ancestry group-specific versions of RVIS (Supplementary Data 1, 2). RVIS regresses the total number of common missense and PTVs on the total number of protein-coding variants in a

gene[1]. The score is then defined as the studentized residual for each gene, with a negative score indicating that a gene is more intolerant to functional variation.

The use of total number of variants on the x-axis in RVIS serves as an empirical proxy for that gene's mutational burden. Here, given that the number of observed variants depends heavily on sample size, we replaced the x-axis with genic mutability estimated from trimer mutation rates (see Methods) (Supplementary Figs. 2, 3). Genic mutability estimates capture the probability of genetic variants occurring within a specific gene, accounting for factors like sequence context, methylation, and gene length. Encouragingly, genic mutability and total number of variants observed per gene were strongly correlated, consistent with our observations in prior work[4] (UKB RVIS: Pearson's $r = 0.93$; gnomAD RVIS: Pearson's $r = 0.94$) (Supplementary Fig. 4a, b).

We next calculated ancestry group-specific versions of RVIS in both the gnomAD and UKB datasets, using mutability on the x-axis and common (MAF > 0.05%) functional variants on the y-axis (Fig. 1a, c). To assess differences in performance of these scores, we compared their

ability to discriminate between genes that have been reported to be associated with severe disease and the rest of the exome. Specifically, we compiled five gene sets: monoallelic developmental and epileptic encephalopathy (DEE) genes ($n = 94$), monoallelic developmental delay (DD) genes ($n = 435$), monoallelic autism spectrum disorder (ASD) genes ($n = 190$), haploinsufficient genes ($n = 390$), and mouse essential genes ($n = 2454$) (Supplementary Data 3).

Across all five gene sets, UKB- and gnomAD-derived RVIS scores that were trained on genetic variation data from the African ancestry cohort consistently achieved the highest area under the ROC curve (Fig. 1b, d). DeLong's test demonstrated that $RVIS_{AFR}$ AUCs were significantly higher than the $RVIS_{NFE}$ AUCs for all gene sets excluding haploinsufficient genes (Supplementary Data 4, 5). As expected, $RVIS_{AMR}$ and $RVIS_{SAS}$ also generally outperformed $RVIS_{NFE}$. A sensitivity analysis demonstrated that this increased performance was robust to different MAF cutoffs for the y-axis (Supplementary Fig. 5). Although the differences in AUC were modest across all, broad genetic representation in population-scale sequencing databases is clearly important in enhancing the resolution of RVIS.

Notably, the AUCs were generally lower for haploinsufficient genes irrespective of ancestry. This is likely because the ClinGen haploinsufficient gene list includes genes associated with a broad spectrum of human disease severities, ranging from phenotypes like dermatitis—which is under relatively modest selective pressure—to those with more substantial health impacts, such as neurodevelopmental disorders. Consistent with this, the distributions of RVIS scores are more variable for haploinsufficient genes compared to neurodevelopmental disorder (NDD) genes for each ancestry group (Supplementary Figs. 6, 7).

We next directly compared the score distributions of $RVIS_{AFR}$ and $RVIS_{NFE}$ for NDD genes (Supplementary Fig. 8). The percentile ranks of these scores were generally well-correlated (Pearson's $r$ range: 0.71–0.85, see Supplementary Fig. 8). However, there were notable outliers that appeared to have discordant intolerance patterns. For example, *SHANK3*, a well-established ASD gene, exhibited a much more intolerant $RVIS_{AFR}$ score compared to $RVIS_{NFE}$ (Supplementary Fig. 8). These results demonstrate how increased genetic diversity can refine estimates of intolerance.

## MTR improves with increasing genetic diversity

Whereas RVIS relies on the observation of common variants to quantify intolerance, other metrics are computed based on the total number of variants observed in a gene—irrespective of allele frequency—compared to a null expectation. Since the total number of observed variants correlates with sample size, we used the UKB to create down-sampled datasets of equal size but with varying compositions of ancestral diversity. Specifically, we defined one cohort of 42,739 exomes ("Maximally Diverse ($n = 43k$)"), which included data from individuals of African ($n = 8701$), South Asian ($n = 9217$), East Asian ($n = 2150$), and Ashkenazi Jewish ($n = 2671$) ancestry, as well as 20,000 randomly sampled non-Finnish European samples (Table 3). As a comparator, we defined a separate cohort of 42,740 NFE individuals ("NFE ($n = 43k$)"). We also included two larger cohorts, including one with all 437,812 NFE samples ("NFE ($n = 440k$)") and a dataset that included all 460,551 of the above samples ("Full Dataset ($n = 460k$)"). The "Maximally Diverse ($n = 43k$)" cohort harbored 3,185,006 missense variants and PTVs compared to 2,132,513 in the "NFE only ($n = 43k$)" cohort (Table 3).

We then constructed gene-level MTR scores using these four cohorts, as well as the ancestry group-specific cohorts (Supplementary Data 7)[9]. MTR is calculated by comparing the observed proportion of missense variants compared to an expected proportion given the sequence context of the gene (see Methods). Across all five gene sets, the "Maximally Diverse"-derived MTR scores achieved the best AUC (Fig. 2). Interestingly, this score also outperformed the "Full Dataset"-

**Table 3 | Total number of unique variants observed in each sub-sampled cohort from the UKB**

| Variant Type | Max Diverse ($n = 43k$) | NFE ($n = 43k$) | NFE ($n = 440k$) | Full Dataset ($n = 460k$) |
|---|---|---|---|---|
| Missense | 2,919,200 | 1,938,545 | 6,101,792 | 7,218,601 |
| PTV | 265,806 | 193,968 | 776,691 | 890,755 |
| Synonymous | 1,492,034 | 945,894 | 2,693,062 | 3,243,081 |

derived score. One possibility is that the overrepresentation of NFE samples in the full dataset dilutes the signal gained from more genetically diverse cohorts. Furthermore, despite the drastic difference in sample size between the down-sampled ($n = 42,740$) and entire ($n = 437,812$) NFE cohorts, the performance of gene-level MTR in these subsets were generally comparable (Supplementary Data 8).

We previously demonstrated that certain genic subregions can be under differential selection to missense variation using a sliding window version of MTR[7,9]. We thus next calculated sliding window MTR for each UKB cohort, employing a 31-codon window (Methods). To compare the performance of these scores, we tested the ability of each to distinguish between ClinVar pathogenic/likely pathogenic variants and control variants. To define control variants, we included variants observed in either gnomAD or TOPMed but not the UKB (Fig. 3a, b). We then compared the performance of these scores to predict de novo variants observed in affected probands versus those observed in their unaffected siblings from denovo-db[20]. AUCs were generally lower for this comparison, as not all de novo variants are necessarily pathogenic. However, this comparison revealed a similar pattern as the ClinVar analysis (Fig. 3c).

Across all three variant sets, the versions of MTR trained on data from the most diverse cohorts achieved the best performance (i.e., the "Maximally Diverse ($n = 43k$)" and the "Full Dataset ($n = 460k$)"). Strikingly, the performance of these two scores was nearly identical, even though the "Maximally Diverse ($n = 43k$)" cohort comprised of roughly $1/10^{th}$ the sample size. The remaining scores seemed to show a greater dependence on sample size than the gene-level metrics. This is likely because the region of interest is smaller for this version of the score (31 codon window versus an entire gene). Exemplifying this, the entire non-Finnish European cohort ($n = 437,812$) outperformed each of the other individual ancestry groups for the three variant sets tested here.

## Loss-of-function intolerance

Finally, we examined the effects of genetic diversity on the performance of LOF intolerance metrics. Specifically, we considered our previously described LOF Observed/Expected metric (LOF O/E) and the related LOF-FDR score[4] (Supplementary Data 9, 10). The LOF O/E score evaluates the ratio of LOF variants against the expected ratio of LOF variants under neutrality. The LOF-FDR score employs a one-sided binomial exact test with Benjamini and Hochberg false discovery rate multiple-testing correction to compare the observed to expected values (Methods). Scores trained with the entire NFE cohort ($n = 440k$) and the complete dataset ($n = 460k$) consistently outperformed the scores trained using reduced non-Finnish European cohort ($n = 42,740$) and the maximally diverse cohort ($n = 42,739$) across all gene sets (Fig. 4, Supplementary Fig. 9). Compared to the prior metrics, sample size generally appeared to be the most important determinant of performance of LOF scores (Supplementary Data 11, 12). The most plausible explanation for this observation is the relative scarcity of protein-truncating variants (PTVs) compared to missense and synonymous variants (Tables 1, 2). PTVs are inherently rarer due to stronger purifying selection against deleterious mutations that disrupt gene function. As a result, even in ancestrally diverse populations, the absolute number of PTVs remains low, and thus larger sample sizes increase the confidence of observed statistics.

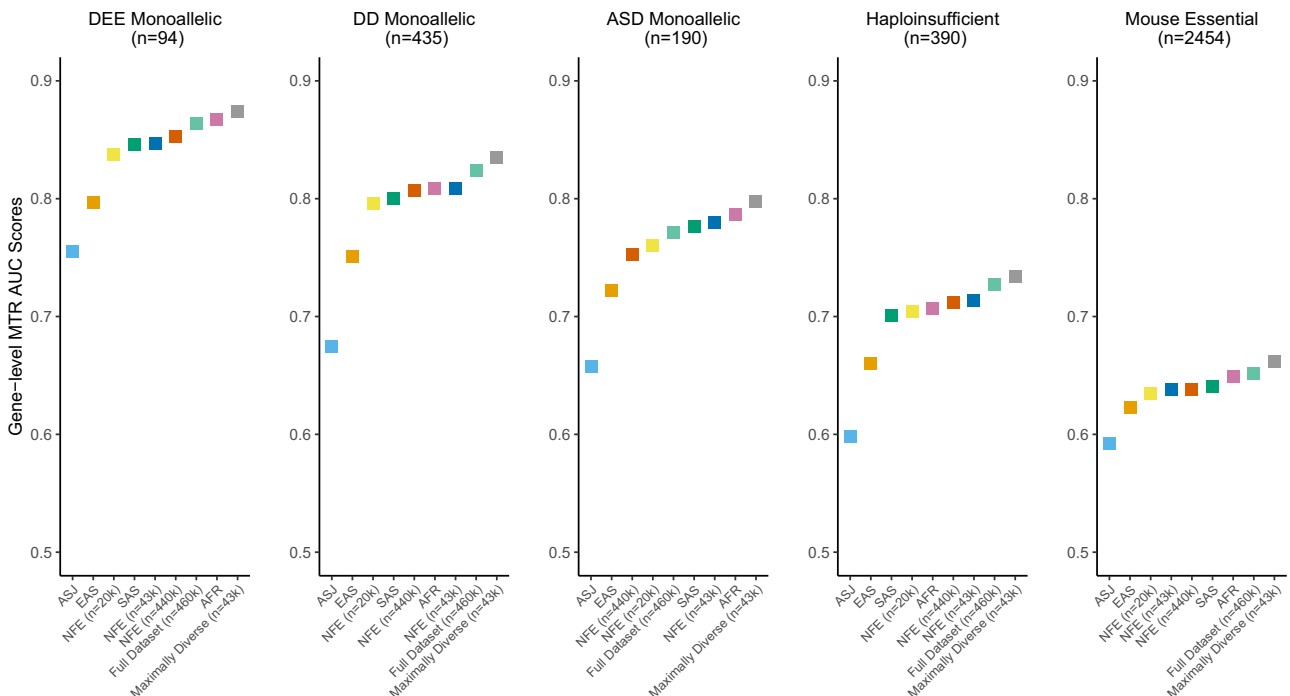

**Fig. 2 | Performance of gene-level MTR scores.** AUC-ROC scores illustrating the ability of each gene-level MTR score to predict five different gene lists. Each score represents a version of the score trained on the pre-defined UKB cohorts. AFR African, ASJ Ashkenazi Jewish, EAS East Asian, SAS South Asian, NFE non-Finnish European, DEE developmental and epileptic encephalopathy, DD developmental delay, ASD autism spectrum disorder.

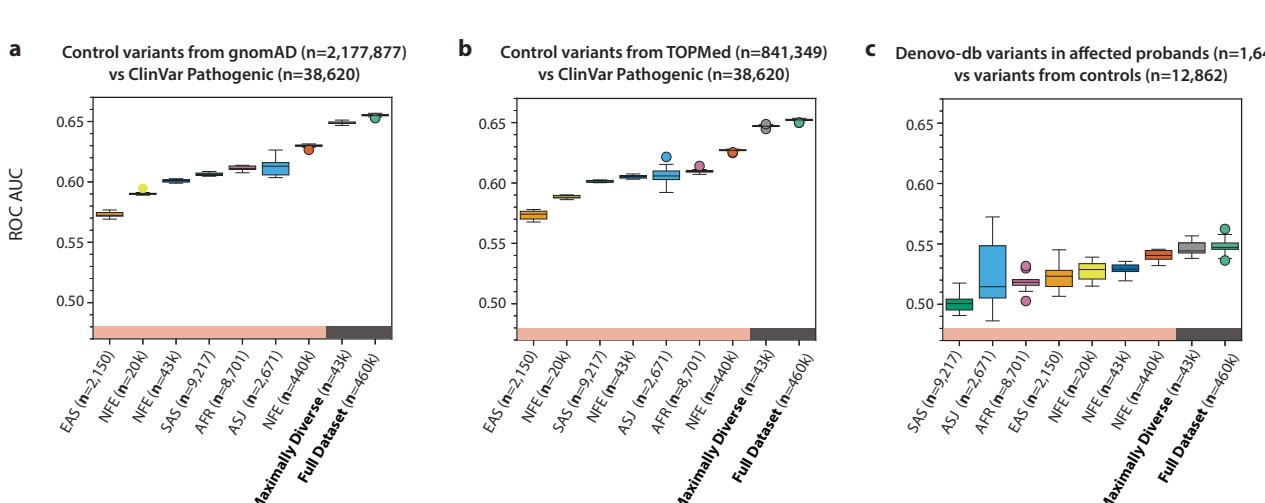

**Fig. 3 | AUC-ROC values of sliding-window MTR in UKB for predicting three variant sets across different ancestry groups. a** AUC-ROC scores reflecting each score's ability to distinguish between ClinVar "pathogenic" or "likely pathogenic" variants and control variants from gnomAD. **b** Same as a, except control variants were derived from TOPMed. **c** Ability of each score to distinguish between de novo variants observed in probands versus unaffected siblings in denovo-db. Control variants from gnomAD are variants found in gnomAD but not in UKB. Control variants from TOPMed are variants found in TOPMed but not in UKB or gnomAD. ClinVar Pathogenic contains ClinVar variants annotated as pathogenic or likely pathogenic. The box plots show the median (centre line), first and third quartiles (box limits), and 1.5x the interquartile range above and below the third and first quartiles (upper and lower whiskers). AFR African, ASJ Ashkenazi Jewish, EAS East Asian, SAS South Asian, NFE non-Finnish European.

## Discussion

Discerning intolerant regions of the human genome is crucial to prioritizing likely disease-causing mutations, facilitating the discovery of new risk genes, and enabling insights into human genome biology more broadly. Here, we performed an extensive study that evaluates the impact of ancestral diversity and sample size have on intolerance metrics using two of the largest population-level exome sequencing databases. Consistent with previous findings, we observed a higher prevalence of common functional variants in non-European ancestry groups compared to European populations. In turn, incorporating

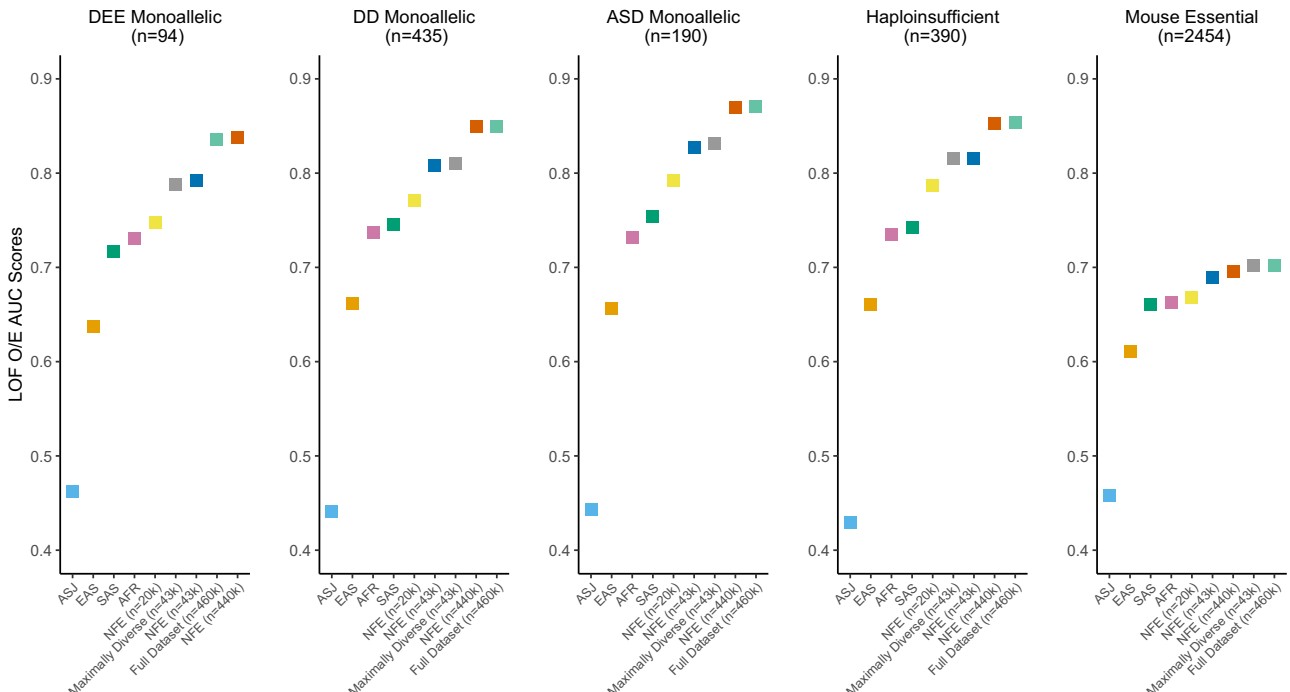

**Fig. 4 | Performance of LOF O/E trained on different UKB cohorts.** AUC-ROC scores illustrating the ability of LOF O/E to predict five different gene lists. Each score represents a version of the score trained on the UKB cohorts composed of different ancestries. AFR African, ASJ Ashkenazi Jewish, EAS East Asian, SAS South Asian, NFE non-Finnish European, DEE developmental and epileptic encephalopathy, DD developmental delay, ASD autism spectrum disorder.

more diverse cohorts demonstrably improved the resolution of various intolerance metrics, which has also been previously observed in non-coding intolerance metrics[10]. These differences were most apparent in RVIS and missense-specific intolerance scores (gene-level MTR and sliding window MTR). In comparison to RVIS and MTR, LOF intolerance metrics seemed more dependent on sample size. This is likely because LOF observations are considerably more sparse than missense and synonymous variants, meaning larger samples are required to achieve meaningful differentiation between test units. Based on improved accuracies across several benchmarks, we recommend users to default to RVIS and missense-specific intolerance scores derived from the African ancestry group and LOF intolerance metric computed from the full dataset. Altogether, our results broadly underscore the importance of achieving broad ancestral representation in sequencing databases.

Our findings have broad implications beyond understanding intolerance to variation in protein coding genes. For example, the FDA's recent guidance on Diversity Action Plans (2024) reinforces the importance of including underrepresented populations in clinical trials to improve the generalizability of study results. Our findings underscore this need by demonstrating that greater ancestral diversity in genetic studies improves intolerance metrics, which could lead to more equitable and effective precision medicine across populations. Secondly, increasing ancestral diversity in genetic datasets could have significant implications for drug discovery. A recent study demonstrated that genetically supported drug targets exhibit higher success rates in clinical trials[21], further emphasizing the importance of incorporating diverse populations in the early stages of therapeutic development. Since the frequency of deleterious and protective alleles may vary across populations, broadening diversity in genomic datasets increases the community's power to identify and de-risk novel targets to move forward in drug development pipelines. Our work also has implications for understanding the transferability of results across populations. In related work, we and others[22–24] have demonstrated that polygenic risk score models trained on European populations often perform poorly when applied to non-European groups, underscoring the importance of diversity in genetic research. The development of intolerance metrics that are more generalizable across different populations could potentially improve the portability of models that combine rare alleles and polygenic risk scores to model disease risk.

Although we analyzed data from several ancestral groups, there still remains incomplete representation of many human populations. For example, the intolerance scores defined from AFR cohorts achieved the best performance, but we currently lack representation from majority of the African continent. In addition, the inclusion of diverse populations that have undergone strong demographic bottlenecks or high levels of consanguinity could also improve intolerance metrics[25]. Because there is an increased rate of homozygous LOFs in these populations, their increased representation would also enable the identification of genes intolerant to recessive variation, which has been notoriously difficult to assess with currently available data. It is also important to note that the human mutation spectrum varies across ancestries[26], and the mutability estimates used here are not ancestry-specific. As sample sizes continue to grow and ancestral representation increases, we recommend reverting to the original RVIS method of using the total number of observed variants on the x-axis. Overall, our work adds to the growing evidence that increasing the ancestral diversity of human genome/exome sequencing datasets is imperative for genomic health equity.

## Methods

### Cohort
The UK Biobank has approval from the North-West Multi-centre Research Ethics Committee (11/NW/0382), and participants provided written informed consent.

### Genetic ancestry groups
Genetic ancestry group is distinct from race and ethnicity as explained by the National Academies of Sciences, Engineering, and Medicine

(NASEM). While race and ethnicity are sociopolitical constructs based on perceived shared biological characteristics and culture, genetic ancestry refers to the specific paths through family trees where DNA is inherited from specific ancestors. Karczewski et al. describe the methodology gnomAD used to determine genetic ancestry group[14]. We used previously published labels for the UKB exomes[17,27].

Splitting these exomes into geographic cohorts is beneficial in this study because genetic variants are observed with varying frequencies between genetic ancestry groups due to demographic histories, including bottleneck events. We want to underscore, however, that the genetic ancestry stratifications in both the UKB and gnomAD are artificially created and not naturally occurring. We recognize the concern behind genetic ancestry group annotations and their potential to exacerbate the illusion of natural ancestral populations as well as the association between geographic labels and socio-politically constructed concepts including race and ethnicity. Considering this and in accordance with recent guidelines from the National Academies of Sciences, Engineering, and Medicine (NASEM), we emphasize that genetic ancestry exists on a spectrum and relates to the family tree where DNA is inherited from specific ancestors. Moreover, genetic ancestry groups are stratified by genetic similarity, and the geographic labels from this study are artificially created.

## Estimating coverage-corrected gene size and filtering variants
We used the gnomAD public transcripts as our coding-sequence source data (gnomAD Predicted Constraint Metrics v2.1.1, GRCh38). The data provided a canonical transcript per gene, defined as the longest isoform. If there were multiple canonical transcripts for the same gene, we retained the transcript with the greatest number of observed variants. Additional predicted constraint metrics including observed and expected variants counts per gene, observed/expected ratio, and Z-scores of the observed counts relative to expected were also included in the data. Mutability of missense, synonymous, and LOF variants per gene were also provided. The mutability metrics were retrieved from the gnomAD metrics file, where the authors used a trinucleotide context model as described in the gnomAD publication[14]. The mutability of LOF variants provided by gnomAD is limited to single nucleotide variants. To calculate the mutability for frameshift variants, we multiplied this mutability rate by 1.25[5]. The sum of these four mutability values was used as the mutability per gene. The UKB variants were annotated with SnpEF[28], and we filtered for the canonical transcript in the gnomAD.

Synonymous variants were identified with the function "synonymous". Missense variants were labeled with the functions "missense", "in frame insertion", and "in frame deletion". Loss-of-function variants were annotated with the functions "frameshift variant", "stop gained", "start lost", "splice acceptor variant", "splice donor variant", "stop gained and frameshift variant", "splice donor variant and coding sequence variant and intron variant", "splice donor variant and intron variant", "stop gained and in frame insertion", "frameshift variant and stop lost", "frameshift variant and start lost", "start lost and splice region variant", "stop gained and protein altering variant", "stop gained and frameshift variant and splice region variant", "stop gained and in frame deletion", and "frameshift variant and stop retained variant".

## Constructing RVIS
RVIS[1] scores were calculated using a cohort of 125,748 exomes from the gnomAD (v2.1) and 460,551 exomes from the UKB. Seven sets of ancestry group-specific RVIS were computed from the gnomAD: AFR ($n = 8128$), SAS ($n = 15,308$), AMR ($n = 17,296$), EAS ($n = 9197$), ASJ ($n = 5040$), NFE ($n = 56,885$), and FIN ($n = 10,824$). Five sets of ancestry group-specific RVIS were derived from the UKB Biobank (UKB): EAS ($n = 2150$), SAS ($n = 9217$), ASJ ($n = 2671$), AFR ($n = 8701$), and NFE ($n = 20k$). The number of common functional variants per given gene

were determined for each ancestry group. These common functional variants were defined as missense and loss-of-function variants with MAF values greater than 0.05%, which was a cutoff threshold validated by our sensitivity analysis (Supplementary Fig. 5). The number of common functional variants were then regressed against mutability, resulting in seven linear regression models for gnomAD and five models for the UKB. The studentized residuals from these models were computed as the ancestry group-specific RVIS.

## Constructing gene-level MTR scores
We generated gene-level MTR scores using the four cohorts and the ancestry group-specific cohorts in the UKB. The formula for computing gene-level MTR was

$$MTR = \frac{\frac{missense(obs)}{missense(obs)+synonymous(obs)}}{\frac{missense(\exp)}{missense(\exp)+synonymous(\exp)}} \quad (1)$$

which compares the ratio of observed fraction of missense variants relative to observed missense and synonymous variants against the expected fraction of missense variants. With the assumption that synonymous variants are evolutionarily neutral and free from selective pressures, the MTR computes the fraction of allowed missense mutations relative to the maximum possible proportion of missense variation in the absence of selective pressure. The gene-level MTR focuses on the cumulative intolerance of the whole gene.

## LOF O/E and LOF-FDR
We generated LOF O/E and LOF-FDR scores using the four cohorts and the ancestry group-specific cohorts in the UKB[4]. Briefly, the expected rate is calculated by taking the mutation rate of LOFs (i.e., PTVs) and dividing that by the sum of mutation rates of all possible mutation effects in the gene, including synonymous, missense, and LOF variants. Mutability estimates were derived from the gnomAD v2.1 metrics file[14]. To account for frameshift variants, we multiplied the PTV mutability by 1.25[5].

$$LOF\ O/E = \frac{\frac{PTV(observed)}{PTV(observed)+Missense(observed)+Synonymous(observed)}}{\frac{PTV(mutability)+PTV(mutability)\cdot 1.25}{PTV(mutability)+PTV(mutability)\cdot 1.25+Synoymous(mutability)+Missense(mutability)}} \quad (2)$$

To calculate LOF-FDR, we computed a one-sided binomial exact test using the number of LOF variants, total number of unique variants, and expected fraction of LOF variants under neutrality and applied Benjamini and Hochberg false discovery rate multiple-testing correction.

## Benchmarking gene-level scores
To test cohort specific gene-level scores in determining disease-causing genes, we performed simple logistic regression models for RVIS, gene-level MTR, LOF O/E, and LOF-FDR (Supplementary Data 4, 5, 8, 11, 12). In these models, the gene-level metric was used as the predictor variable, while a binary variable distinguishing disease-causing genes from the rest of the HGNC genes served as the response variable. For each pairing of test set and gene-level score, the area under the ROC curve (AUC) for identifying disease causing genes was computed. DeLong's test was used to assess for significant difference in the predictive ability between models.

Five gene sets were used as test sets to evaluate performance of the cohort specific gene-level scores. These were monoallelic DEE genes, monoallelic developmental delay (DD) genes, monoallelic autism spectrum disorder (ASD) genes[29], haploinsufficient genes[30], and mouse essential genes[31]. The monoallelic ASD gene list was curated by selecting SFARI Tier 1 ASD genes ($n = 207$) and filtering genes with biallelic mechanism, leaving 190 total monoallelic ASD genes. The monoallelic DD gene list was determined by selecting genes with

"Brain/Cognition" and "Definitive" labels from the Developmental Disorder Genotype-Phenotype Database (DD2GP) then filtering for monoallelic genes ($n = 218$) and combining these genes with 199 genes found to be significant from the Kaplanis et al, culminating in total of 435 genes for the DD monoallelic set[32]. The monoallelic DEE gene list was created by selecting for "DEE" phenotype genes from Online Mendelian Inheritance in Man (OMIM) and combining these with previously curated gene lists[29,33–37]. We provided additional information on curating monoallelic DEE, DD, and ASD gene lists in our previous study[29].

### Sliding window MTR variant-level benchmark

Sliding window MTR scores were calculated as described in Traynelis et al. and Vitsios et al. for every codon position of 18,823 canonical transcripts over a sliding window of 31 codons[7,9]. We employed variant-level data from nine different cohorts (of single or diverse ancestry groups) derived from the UK Biobank (UKB): EAS ($n = 2150$), SAS ($n = 9217$), ASJ ($n = 2671$), AFR ($n = 8701$), NFE ($n = 20$k), NFE ($n = 43$k), NFE ($n = 440$k), Maximally Diverse ($n = 43$k) and Full Dataset ($n = 460$k). To test MTR's ability to discriminate pathogenic missense variants from benign, we composed three sets of benign variants and two sets of pathogenic variants. The benign sets we compiled comprise of: i) variants present in gnomAD (v2.1), but not present in UKB, ii) variants present in TOPMed, but not present in UKB or gnomAD (v2.1), and iii) de novo variants from denovo-db present in control individuals from the SSC or non-SSC samples[20]. As pathogenic variant sets, we used: (i) variants confidently annotated in ClinVar as "Pathogenic" or "Likely pathogenic", or (ii) de novo variants from denovo-db present in affected individuals from the SSC or non-SSC samples.

To test MTR's performance, as calculated across the different cohorts, in distinguishing pathogenic missense variants form benign, we used MTR as a measure of probability that a variant is benign: the higher the MTR score, the greater the probability. For each combination of test sets and MTR metrics, the area under the ROC curve (AUC) for distinguishing pathogenic from benign variants was calculated. Due to the imbalance in the respective positive and negative test sets, we randomly down-sampled the larger one in each case to match the size of the other set of the pair. Finally, we repeated the down-sampling and calculated AUC scores across 10 iterations for each pair of pathogenic-benign variant sets to include enough data points and increase the robustness of the discrimination tasks.

### Reporting summary

Further information on research design is available in the Nature Portfolio Reporting Summary linked to this article.

## Data availability

The Gene-level intolerance scores generated in this study are included in supplementary data, and the sliding window MTR scores are publicly available on FigShare (https://doi.org/10.6084/m9.figshare.26049661. v1)[38]. Scores are also browsable via our publicly available portal: http://intolerance.public.cgr.astrazeneca.com/. The UK Biobank whole-exome sequencing data are publicly available to registered researchers through the UKB data access protocol. Exomes can be found in the UKB showcase portal: https://biobank.ndph.ox.ac.uk/showcase/label. cgi?id=170. Additional information about registration for access to the data is available at http://www.ukbiobank.ac.uk/register-apply/. Data for this study were obtained under Resource Application Number 26041 and 65851. gnomAD v2.1.1 data are publicly available through the gnomAD website (https://gnomad.broadinstitute.org/data#v2). Gene lists are available through GitHub (https://github.com/alhanster/ancestral_diversity/tree/main/data/genelist); denovo-db v1.6.1 data is publicly available through https://denovo-db.gs.washington.edu/denovo-db/; and TOPMed freeze 5 data are publicly available through https://bravo.sph.umich.edu.

## Code availability

Code for generating gene-level scores is available at: https://github.com/alhanster/ancestral_diversity and the code for generating sliding-window MTR scores is available at: https://github.com/astrazeneca-cgr-publications/OncMTR/tree/mtr-ancestry-specific. Code is also available via Zenodo (https://zenodo.org/records/14901599)[39].

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

## Acknowledgements

We thank the participants in the UK Biobank and gnomAD for their contributions to research (UKB Resource Application Numbers 26041 and 65851). We extend our gratitude to the AstraZeneca Centre for Genomics Research Analytics and Informatics team for processing and analysis of sequencing data. R.S.D. is supported by a Longevity Impetus Grant from Norn Group, Hevolution Foundation, and Rosenkranz Foundation.

## Author contributions

R.S.D. and S.P. designed the study. A.L.H., D.M., D.V., and R.S.D. performed analyses and statistical interpretation. Q.W. and F.H. performed bioinformatics processing. A.L.H., C.F.S., D.V., J.C.B., D.M., and R.S.D. performed data visualization. D.M. and D.V. developed the web portal. A.L.H., C.F.S., and R.S.D. wrote the manuscript. A.L.H., C.F.S., D.M., J.C.B., V.R., F.H., E.V.G., N.K., C.D.B., Q.W., S.P., D.V., and R.S.D. reviewed the manuscript.

## Competing interests

D.M., F.H., Q.W., S.P., and D.V. are current employees and/or stockholders of AstraZeneca. R.S.D. is a paid consultant of AstraZeneca. N.K. and C.D.B. are employees of Galatea Bio. The remaining authors declare no competing interests.
