## [Transparent Peer Review file · Nature Communications]

Diverse ancestral representation improves genetic intolerance metrics

Corresponding Author: Dr Ryan Dhindsa

Version 0:

Reviewer comments:

Reviewer #1

(Remarks to the Author)

Han et al assessed how genetic intolerance metrics performed under different database compositions. This included the use of multiple large-scale genetic databases (UK Biobank and gnomAD) with varying compositions of genetic ancestries. Han et al found that most genetic intolerance metrics performed better when the dataset was composed of multiple genetic ancestries. Other metrics seemed to be more impacted by sample size than sample composition. This manuscript has important implications for the future use of these metrics due to most genetic data coming from individuals who are mostly of European ancestry. This paper adds to the growing body of literature that stresses the need for increasing the ancestral diversity of human genome/exome sequencing datasets. I think the conclusions are well supported by their analyses.

Major comments:

1. There are a lot of abbreviations throughout the text that might make this manuscript difficult to understand for a general reader that is not familiar with these metrics. I would suggest reducing the number of abbreviations or adding a glossary to the supplement or at the end of the main text. There are also a few abbreviations that are not defined at their first appearance in the main text:
 - a. Line 61: "LOF"
 - b. Line 99: "NFE"
 - c. Line 100: "PTVs"
2. Lines 108 – 111: How many of the new rare variants added with increasing sample size are singletons vs non-singletons. Is the signal of not being saturated for rare variants primarily being driven by singletons being added or is this observed across all frequencies of rare variants?
3. Line 143-145: "We next calculated ancestry group-specific versions of RVIS in both the gnomAD and UKB datasets, using mutability for the x-axis and common (MAF > 0.05%) functional variants on the y-axis (Fig 2A, B)." Should this be referencing figure 1 A,C instead of 2A,B?
4. Line 278-280: "In addition, the inclusion of diverse populations that have undergone strong demographic bottlenecks or high levels of consanguinity could also improve intolerance metrics" Are you able to comment on this with your dataset that includes ~10,000 Finish samples?

Minor comments:

1. TOPMed has two versions in the text: 1) TopMED (Line 214 and Figure 3 caption) and 2) TOPMed (Line 394). I think "TOPMed" is the correct version.
2. Is the presence/absence of a variant in TOPMed meta-data that is recorded in gnomAD/UK Biobank? I wasn't sure how this was determined, and I also couldn't find TOPMed in the data availability section.
3. Line 373-374: What are the numbers in parentheses referencing: "haploinsufficient genes (26014595), and mouse essential genes (21051359, 23675308, 23843252)."
4. Line 154-156: "DeLong's test demonstrated that RVISAFR AUCs were significantly higher than the RVISNFE AUCs for all gene sets excluding haploinsufficient genes (Supplementary Table 4,5)." I noticed the RVISAFR did have a $P < 0.05$ for some of the other ancestral groups compared (rv_{is_asj}, rv_{is_eas}, rv_{is_sas}, rv_{is_fin}). Is there a potential explanation for why the RVIS_AFR performs significantly better than these ancestry groups but not the NFE group specifically for the haploinsufficient genes?

(Remarks on code availability)

The code is made available with a detailed README file that provides detailed instructions on how to run the code.

Reviewer #2

(Remarks to the Author)

Han and colleagues have presented a research demonstrating that inclusion of diverse populations improves the power of genetic intolerance metrics. Given the emerging value of developing genetic intolerance metrics using large-scale human population data, this manuscript develops ancestry-specific intolerance metrics at gene and sub-genic level. The ancestry-specific constraint metrics are benchmarked by classification of disease genes and variants from case and control. I have the following major concerns:

1. It's known that mutation spectrum is population-specific (e.g. Kelly Harris and others showed before <https://elifesciences.org/articles/24284>).For example, the transition 5'-TCC-3' to 5'-TTC-3' is enriched in European populations compared to Africans. Thus I would imagine the genic mutability deriving from the trimer mutation rate might also have population difference. I also suspect the improved power by using a specific population shall be reflected by its higher ancestry-specific mutability. Have the authors observed this? It's not clear how the genic mutability used in the manuscript is derived and whether the authors have considered the population-specific mutation rate and genic mutability. If there is indeed population-specific genic mutability, the authors shall also update their population-specific genetic intolerance estimates throughout the manuscript.
2. To assess the performance of ancestry-specific RVIS score, the authors compared their AUCs in classifying disease genes. I think it would be helpful to add the visualisation on comparing the distribution of ancestry-specific RVIS scores. This might help to understand why some ancestry-specific RVISs have improved the classification statistically. I imagine there are some genes have improved statistical power by using ancestry-specific RVIS compared to the NFE one and there are large overlap and also some discrepancies between different ancestry-specific RVIS. Can the authors demonstrate this?

(Remarks on code availability)

There are instructions provided to reproduce the figures and scripts to generate the scores.

Version 1:

Reviewer comments:

Reviewer #1

(Remarks to the Author)

In this revised manuscript, Han et al sufficiently addressed all of my questions and comments. I agree with the authors that developing a new recessive intolerance metric is beyond the scope of this paper, and I appreciate them further highlighting this importance for future research. I do not have any new major comments for the authors to address. I only have one minor comment.

1. Could you add "NDD" (on line 172 and 174) to the glossary and also define this at the first mention of "neurodevelopmental disorders." I think the first occurrence is on line 171.

(Remarks on code availability)

Both github repositories have a useful readme detailing how to run the code.

Reviewer #2

(Remarks to the Author)

The authors have addressed the majority of my comments except for comment 1:

I agree that the total number variants would be an empirical proxy for mutability and sample size given different ancestral groups. The logic about using the mutability as the X-axis is not clear(Line 143-146):

The use of total number of variants on the X-axis in RVIS serves as an empirical proxy for that gene's mutability and its length. Here, given that the number of observed variants depends heavily on sample size, we replaced the X-axis with genic mutability estimated from trimer mutation rates (see Methods).

I believe the first sentence shall goes like: The use of total number of variants on the X-axis in RVIS serves as an empirical proxy for that gene's mutability and sample size. I suggest also providing definition of gene's mutability here. As gene's mutability has already accounted for length. For the second sentence the author explained using gene mutability as X-axis due to concern about sample size. But as the Y-axis (total number of common variants) also depends on the sample size, using the total number of observed variants is more reasonable as it would regress sample size out. As I understand in the case of ignoring variations of mutability across ancestries here, comparison across genes fairly within ancestry groups is what matters most here so concern about sample size doesn't sound like the reason for using mutability as X-axis. I suggest the authors could update their reasoning on the model design and make it clear to general readers.

I have additional minor comments - if a gene has two opposite intolerance classifications across ancestral groups, what would you suggest users do?

(Remarks on code availability)

The repository has provided the key scripts to reproduce the analysis.

Version 2:

Reviewer comments:

Reviewer #2

(Remarks to the Author)

The authors have addressed all of my comments.

(Remarks on code availability)

The authors have provided essential scripts to reproduce the analysis.

We want to thank the editorial team and reviewers for their constructive comments and the opportunity to improve our manuscript. In particular, we would like to highlight some key areas of improvement:

- 1) *In response to the reviewers' comments, we have examined whether the newly detected rare variants with increasing sample size are driven by singletons. This further supports our initial assertion that we have saturated the discovery of more common variants for many ancestries, and that as sample sizes grow, we will preferentially continue to identify variants at the rarer end of the allele frequency spectrum.*
- 2) *We identified NDD genes with discordant RVIS percentile ranks between ancestries, drawing attention to potential genes that may have more accurate estimates derived from non-European ancestries.*
- 3) *We have included several new visualizations to show differences in ancestry-specific scores across gene lists. Importantly, we now include a publicly available portal (<http://intolerance.public.cgr.astrazeneca.com>) that allows users to query any protein-coding gene and visualize all of the ancestry-specific intolerance metrics.*
- 4) *Finally, we demonstrate that total number of observed variants heavily depends on sample size regardless of ancestry, providing further support for replacing X-axis with mutability when there is a large imbalance of genetic ancestries adopted for training.*

Overall, we believe the aforementioned changes as well as incorporating other reviewer suggestions for readability have substantially improved our paper. We provide a point-by-point response to reviewer comments below.

REVIEWER COMMENTS

Reviewer #1

Han et al assessed how genetic intolerance metrics performed under different database compositions. This included the use of multiple large-scale genetic databases (UK Biobank and gnomAD) with varying compositions of genetic ancestries. Han et al found that most genetic intolerance metrics performed better when the dataset was composed of multiple genetic ancestries. Other metrics seemed to be more impacted by sample size than sample composition. This manuscript has important implications for the future use of these metrics due to most genetic data coming from individuals who are mostly of European ancestry. This paper adds to the growing body of literature that stresses the need for increasing the ancestral diversity of human genome/exome sequencing datasets. I think the conclusions are well supported by their analyses.

Thank you for your thoughtful review and for recognizing the significance of our findings. We are especially pleased that you see this work as contributing to the growing evidence for the need to increase representation in human genetics. We also appreciate your detailed comments and suggestions, which we address below.

Major Comments:

1. There are a lot of abbreviations throughout the text that might make this manuscript difficult to understand for a general reader that is not familiar with these metrics. I would suggest reducing the number of abbreviations or adding a glossary to the supplement or at the end of the main text. There are also a few abbreviations that are not defined at their first appearance in the main text: a. Line 61: “LOF” b. Line 99: “NFE” c. Line 100: “PTVs”

We have now added a glossary that defines these abbreviations as Supplementary Table 1.

2. Lines 108 – 111: How many of the new rare variants added with increasing sample size are singletons vs non-singletons. Is the signal of not being saturated for rare variants primarily being driven by singletons being added or is this observed across all frequencies of rare variants?

We compared the proportion of singletons among rare variants (MAF < 0.05%) across the NFE 20k, 43k, and 440k subsets. In all subsets, the majority (>50%) of newly added variants were singletons. As expected, protein-truncating variants (PTVs) showed a relatively higher proportion of singletons compared to missense and synonymous variants, reflecting stronger selection on this class of variation. We now include this plot as Supplementary Figure 1.

3. Line 143-145: “We next calculated ancestry group-specific versions of RVIS in both the gnomAD and UKB datasets, using mutability for the x-axis and common (MAF > 0.05%) functional variants on the y-axis (Fig 2A, B).” Should this be referencing figure 1 A,C instead of 2A,B?

We have revised this accordingly.

4. Line 278-280: “In addition, the inclusion of diverse populations that have undergone strong demographic bottlenecks or high levels of consanguinity could also improve intolerance metrics” Are you able to comment on this with your dataset that includes ~10,000 Finish samples?

Thank you for highlighting this point. We originally intended to emphasize that the inclusion of populations with strong demographic bottlenecks or high levels of consanguinity—such as the approximately 10,000 Finnish samples in our dataset—could be especially important for

developing recessive intolerance metrics, which remains a major challenge in the field. Critically, there is currently a lack of comprehensive methodologies for assessing recessive genetic intolerance. In our current study, we intentionally focused on examining well-established intolerance scores that predominantly capture dominant variation. Developing a new recessive intolerance metric would go beyond the scope of this paper.

In our revised manuscript, we have clarified the outstanding need for recessive intolerance metrics and highlighted the importance of incorporating populations with strong demographic bottlenecks or high levels of consanguinity in future research aimed at creating these metrics.

Minor comments:

1. TOPMed has two versions in the text: 1) TopMED (Line 214 and Figure 3 caption) and 2) TOPMed (Line 394). I think “TOPMed” is the correct version.

We have addressed this accordingly.

2. Is the presence/absence of a variant in TOPMed meta-data that is recorded in gnomAD/UK Biobank? I wasn't sure how this was determined, and I also couldn't find TOPMed in the data availability section.

Thank you for bringing this to our attention. The presence or absence of a variant in TOPMed is not recorded in the metadata of gnomAD or UK Biobank. Instead, to determine the variants unique to TOPMed, we downloaded the TOPMed VCF files and intersected them with the variants from gnomAD and UK Biobank. This allowed us to identify variants present in TOPMed but absent in the other two datasets. We have clarified this on lines 232-233. Additionally, we have added the TOPMed information to the Data Availability section.

3. Line 373-374: What are the numbers in parentheses referencing: “haploinsufficient genes (26014595), and mouse essential genes (21051359, 23675308, 23843252).”

Thank you for pointing this out. These were PMIDs corresponding to the gene lists. We have removed them in our revision and replaced them with appropriate citations.

4. Line 154-156: “DeLong’s test demonstrated that RVISAFR AUCs were significantly higher than the RVISNFE AUCs for all gene sets excluding haploinsufficient genes (Supplementary Table 4,5).” I noticed the RVISAFR did have a $P < 0.05$ for some of the other ancestral groups compared (rvis_asj, rvis_eas, rvis_sas, rvis_fin). Is there a potential explanation for why the RVIS_AFR performs significantly better than these ancestry groups but not the NFE group specifically for the haploinsufficient genes?

We believe that RVIS_{AFR} performs significantly better than other ancestry groups—but not the NFE group—for haploinsufficient genes due to a wider spectrum of disease severity captured within this gene set. According to ClinGen’s definition, haploinsufficient genes are those where loss-of-function mutations consistently cause the same phenotype, regardless of severity (https://clinicalgenome.org/site/assets/files/6428/dosage_sop-scoring-1.pdf). This means some haploinsufficient genes are highly intolerant because they cause severe phenotypes that reduce reproductive fitness (e.g., **SCN1A** associated with epileptic encephalopathy), while others are linked to more benign conditions under weaker selection pressure (e.g., **FLG** associated with skin conditions).

There is more variance in intolerance scores among the full list of haploinsufficient genes compared to neurodevelopmental disease-specific gene lists (included as Supplementary Figures 5 and 6 in the revised manuscript).
(Supplementary Figures 5, 6).

Reviewer #1 (Remarks on code availability):

The code is made available with a detailed README file that provides detailed instructions on how to run the code.

Reviewer #2

Han and colleagues have presented a research demonstrating that inclusion of diverse populations improves the power of genetic intolerance metrics. Given the emerging value of developing genetic intolerance metrics using large-scale human population data, this manuscript develops ancestry-specific intolerance metrics at gene and sub-genic level. The ancestry-specific constraint metrics are benchmarked by classification of disease genes and variants from case and control.

Major Comments:

1. It's known that mutation spectrum is population-specific (e.g. Kelly Harris and others showed before <https://elifesciences.org/articles/24284>). For example, the transition 5'-TCC-3' to 5'-TTC-3' is enriched in European populations compared to Africans. Thus I would imagine the genic mutability deriving from the trimer mutation rate might also have population difference. I also suspect the improved power by using a specific population shall be reflected by its higher ancestry-specific mutability. Have the authors observed this? It's not clear how the genic mutability used in the manuscript is derived and whether the authors have considered the population-specific mutation rate and genic mutability. If there is indeed population-specific genic mutability, the authors shall also update their population-specific genetic intolerance estimates throughout the manuscript.

Thank you for highlighting this point. To the best of our knowledge, well-validated ancestry-specific mutability estimates are not currently available for all the ancestral groups included in our study. Therefore, we opted to use the mutability estimates from gnomAD (Karczewski et al., 2020), which are calculated based on a multi-ancestry cohort. While these estimates are not ancestry-specific, they incorporate data from individuals across the major ancestral groups observed in gnomAD.

We have revised the Methods section (Lines 360–365) to provide a clearer explanation of how genic mutability was derived in our study. We also now acknowledge the limitation of not incorporating population-specific mutation rates in the discussion (Lines 324–328).

Additionally, we note that the original RVIS methodology (Petrovski et al., 2013) used the total number of observed variants per gene as an empirical estimate of observed mutability, rather than relying on theoretical mutability estimates. If sample sizes were equal across ancestry groups, using the original RVIS formulation would—by design—account for differences in mutation rates between populations. In our revision, we quantified the differences in the total number of observed variants per gene across different cohorts, as illustrated in Supplementary Figures 2 and 3 (also included below).

As expected, we found that the total number of observed variants per gene is heavily influenced by sample size, regardless of ancestry (Supplementary Figures 2 and 3). Therefore, once sample sizes become more equitable across ancestries, we recommend reverting to the original RVIS methodology, which will inherently account for differences in mutation rates between groups. We have included this recommendation in the revised Discussion.

2. To assess the performance of ancestry-specific RVIS score, the authors compared their AUCs in classifying disease genes. I think it would be helpful to add the visualisation on comparing the distribution of ancestry-specific RVIS scores. This might help to understand why some ancestry-specific RVISs have improved the classification statistically. I imagine there are some genes have improved statistical power by using ancestry-specific RVIS compared to the NFE one and there are large overlap and also some discrepancies between different ancestry-specific RVIS. Can the authors demonstrate this?

Thank you for this suggestion. We have incorporated two new additions to the manuscript to help visualize differences in score distributions:

- 1) We have now created a public portal (<http://intolerance.public.cgr.astrazeneca.com>) that allows users to query any protein-coding gene and visualize all of the ancestry-specific intolerance metrics available in the manuscript. We also include a simple schematic showing where that gene falls within the distribution of intolerance scores for each ancestry, as well as its percentile rank (see example below for RVIS)

DNM1

dynamin 1

RVIS (UKB)

Residual variation intolerance score (RVIS) ranks genes according to whether they have more or less common functional variants relative to expectation. Negative scores are likely to reflect purifying selection, whereas positive scores are likely to reflect either absence of purifying selection, presence of some form of balanced or positive selection, or both.

Ancestry [▲]	Value	Percentile	Distribution	Prediction
AFR	-1.323	2.4		⚡
ASJ	-1.303	2.5		⚡
EAS	-0.398	24.8		
NFE	-0.814	7.1		
SAS	-1.449	2.3		⚡

- 2) We have added a figure that compares the differences in percentile ranks of ancestry group-specific RVIS scores for each of the three NDD Gene lists (**Supplementary Figures 5 and 6; see also below**). While the scores for these genes were generally correlated, there were some notable differences in scores across ancestries. For example, *SHANK3*, a well-established ASD gene, exhibited a much more intolerant $RVIS_{AFR}$ score compared to $RVIS_{NFE}$. Other notable examples include *SMO* and *MECP2*, which were much more intolerant in $RVIS_{AFR}$ than $RVIS_{NFE}$, with differences larger than 50%.

Reviewer #2 (Remarks on code availability):

There are instructions provided to reproduce the figures and scripts to generate the scores.

We want to thank the editorial team and reviewers for their constructive comments and the opportunity to improve our manuscript. We provide a point-by-point response to reviewer comments below.

REVIEWER COMMENTS

Reviewer #1

In this revised manuscript, Han et al sufficiently addressed all of my questions and comments. I agree with the authors that developing a new recessive intolerance metric is beyond the scope of this paper, and I appreciate them further highlighting this importance for future research. I do not have any new major comments for the authors to address. I only have one minor comment.

We thank you the reviewer for their thoughtful review.

1. Could you add "NDD" (on line 172 and 174) to the glossary and also define this at the first mention of "neurodevelopmental disorders." I think the first occurrence is on line 171.

Thank you for bringing this to our attention. We have now added "NDD" to the glossary and defined the abbreviation.

Reviewer #1 (Remarks on code availability):

Both github repositories have a useful readme detailing how to run the code.

Reviewer #2

The authors have addressed the majority of my comments except for comment 1:

I agree that the total number variants would be an empirical proxy for mutability and sample size given different ancestral groups. The logic about using the mutability as the X-axis is not clear (Line 143-146):

The use of total number of variants on the X-axis in RVIS serves as an empirical proxy for that gene's mutability and its length. Here, given that the number of observed variants depends heavily on sample size, we replaced the X-axis with genic mutability estimated from trimer mutation rates (see Methods).

I believe the first sentence shall goes like: The use of total number of variants on the X-axis in RVIS serves as an empirical proxy for that gene's mutability and sample size. I suggest also providing definition of gene's mutability here. As gene's mutability has already accounted for length.

For the second sentence the author explained using gene mutability as X-axis due to concern about sample size. But as the Y-axis (total number of common variants) also depends on the sample size, using the total number of observed variants is more reasonable as it would regress sample size out. As I understand in the case of ignoring variations of mutability across ancestries here, comparison across genes fairly within ancestry groups is what matters most here so concern about sample size doesn't sound like the reason for using mutability as X-axis. I suggest the authors could update their reasoning on the model design and make it clear to general readers.

We appreciate the suggestion to better explain the reason behind our decision to use mutability in calculating RVIS. The reviewer is correct that mutability accounts for gene length; we have updated the language on lines and included a definition of genic mutability (lines 143-147).

In Table 2 of our manuscript, we demonstrated that the number of common variants remains constant across increasing sample sizes of the NFE cohort. On the other hand, the number of rare variants increased with sample size. This is consistent with population genetic theory that the number of detectable variants above a certain allele frequency in the population will remain consistent as more samples are added. However, this is not true for rare variants, especially (private) singletons. For every new sample added to the cohort, the sum of the total observed variants in the cohort (of any allele frequency) will continue to grow (Table 2, Supplementary Fig 1). These observations motivated us to adopt mutability as our X-axis rather than total number of observed variants, given the differences in cohort sample sizes between ancestral groups. We also now highlight that the mutability-based RVIS is an established formulation (Petrovski et al., 2015). To avoid sample size being a potential driver of diverging signals, we determined that using mutability would be more conservative in comparing the different ancestries that have varying sample sizes.

I have additional minor comments - if a gene has two opposite intolerance classifications across ancestral groups, what would you suggest users do?

Thank you for highlighting this point. Based on its improved accuracy when measured across several benchmarks, we recommend users prioritise the statistics generated from the African ancestry group, and we include a comment on this in the discussion (lines 299-301). As sample size and ancestral representation continues to improve toward parity, we envision ancestry-specific intolerance scores to eventually be used in diagnostic settings.

Reviewer #2 (Remarks on code availability):

The repository has provided the key scripts to reproduce the analysis.